# Harnessing Gene Expression Profiles for the Identification of Ex Vivo Drug Response Genes in Pediatric Acute Myeloid Leukemia

**DOI:** 10.3390/cancers12051247

**Published:** 2020-05-15

**Authors:** David G.J. Cucchi, Costa Bachas, Marry M. van den Heuvel-Eibrink, Susan T.C.J.M. Arentsen-Peters, Zinia J. Kwidama, Gerrit J. Schuurhuis, Yehuda G. Assaraf, Valérie de Haas, Gertjan J.L. Kaspers, Jacqueline Cloos

**Affiliations:** 1Hematology, Cancer Center Amsterdam, Amsterdam UMC, Vrije Universiteit Amsterdam, 1081 HV Amsterdam, The Netherlands; d.cucchi@amsterdamumc.nl (D.G.J.C.); c.bachas@amsterdamumc.nl (C.B.); zj.kwidama@amsterdamumc.nl (Z.J.K.); gerritjanschuurhuis@gmail.com (G.J.S.); 2Department of Pediatric Oncology/Hematology, Erasmus MC–Sophia Children’s Hospital, 3015 CN Rotterdam, The Netherlands; m.m.vandenheuvel-eibrink@prinsesmaximacentrum.nl; 3Princess Máxima Center for Pediatric Oncology, 3584 CS Utrecht, The Netherlands; T.C.J.Peters-3@prinsesmaximacentrum.nl (S.T.C.J.M.A.-P.); vdehaas@skion.nl (V.d.H.); GJL.Kaspers@amsterdamumc.nl (G.J.L.K.); 4The Fred Wyszkowski Cancer Research, Laboratory, Department of Biology, Technion-Israel Institute of Technology, 3200003 Haifa, Israel; assaraf@technion.ac.il; 5Emma’s Children’s Hospital, Amsterdam UMC, Vrije Universiteit Amsterdam, Pediatric Oncology, 1081 HV Amsterdam, The Netherlands

**Keywords:** pediatric acute myeloid leukemia, drug resistance, drug response, gene expression, chemotherapy, cytarabine, daunorubicin, cladribine, etoposide

## Abstract

Novel treatment strategies are of paramount importance to improve clinical outcomes in pediatric AML. Since chemotherapy is likely to remain the cornerstone of curative treatment of AML, insights in the molecular mechanisms that determine its cytotoxic effects could aid further treatment optimization. To assess which genes and pathways are implicated in tumor drug resistance, we correlated ex vivo drug response data to genome-wide gene expression profiles of 73 primary pediatric AML samples obtained at initial diagnosis. Ex vivo response of primary AML blasts towards cytarabine (Ara C), daunorubicin (DNR), etoposide (VP16), and cladribine (2-CdA) was associated with the expression of 101, 345, 206, and 599 genes, respectively (*p* < 0.001, FDR 0.004–0.416). Microarray based expression of multiple genes was technically validated using qRT-PCR for a selection of genes. Moreover, expression levels of *BRE*, *HIF1A*, and *CLEC7A* were confirmed to be significantly (*p* < 0.05) associated with ex vivo drug response in an independent set of 48 primary pediatric AML patients. We present unique data that addresses transcriptomic analyses of the mechanisms underlying ex vivo drug response of primary tumor samples. Our data suggest that distinct gene expression profiles are associated with ex vivo drug response, and may confer a priori drug resistance in leukemic cells. The described associations represent a fundament for the development of interventions to overcome drug resistance in AML, and maximize the benefits of current chemotherapy for sensitive patients.

## 1. Introduction

The vast majority of pediatric acute myeloid leukemia (AML) patients achieves complete remission with current intensive chemotherapy protocols [1]. Despite optimal upfront therapy, 30–40% of patients relapse [2], and this substantial fraction of patients has not decreased with the introduction of newer treatment protocols [3]. Although 60–70% of these patients achieve second complete remission [4], salvage therapy often results in short- and long-term side effects. Therefore, reducing the requirement for salvage therapy is desirable. Several approaches to prevent relapse may be effective. One could focus on suppression of resistant clones harboring targetable mutations that may have been selected during induction chemotherapy [5], e.g., using FLT3 or IDH1/2 inhibitors [6,7], which requires additional monitoring of clonal heterogeneity at diagnosis and in complete remission. An alternative approach to prevent relapse is the eradication of all leukemic cells with initial (combination) therapy. 

The backbone of chemotherapeutic regimens in AML contains the deoxynucleoside analogue cytosine arabinoside (Ara C), combined with an anthracycline such as daunorubicin (DNR), and often with the topoisomerase inhibitor etoposide (VP16) [8]. In adult AML patients, ex vivo resistance towards Ara C is associated with an increased risk of relapse and poor outcomes, mediated by blast-specific or stroma-specific phenotypes [9,10]. However, controversy exists with regard to the translational capacity of primary AML samples for ex vivo drug sensitivity testing [11,12,13,14,15]. Ex vivo culture data are hampered by variation in cell viability and limited cell proliferation. We have developed strict protocols and quality control to ensure reliable results for ex vivo drug testing of primary leukemia samples. Using highly standardized assays, data on ex vivo drug response have been successfully linked to resistance mechanisms that were shown to be relevant in further studies [16,17].

Some previous studies reported that patient samples with a high ex vivo drug resistance towards Ara C were not only resistant towards other deoxynucleotide analogues (e.g., fludarabine or gemcitabine) [18], but also towards other chemotherapeutics with different modes of action (such as VP16 or DNR) [19,20]. An intrinsic property of leukemic cells that is thought to play an important role in resistance to Ara C, is the ability to transport the drug across the plasma membrane in both directions [21]. The influx of Ara C may be influenced by the expression of nucleoside transporter proteins that limit drug accumulation at a standard dose of Ara C, e.g., via the equilibrate nucleoside transporter protein (*hENT1/SLC29A1*) [22,23]. Additional mechanisms of Ara C resistance concern altered oxidative phosphorylation status [24], *NT5C2* mutations and alteration of *DCK* and *SAMHD1* expression levels [25,26,27]. On the other hand, the effectiveness of other drugs may also be hampered by drug efflux, which is strongly influenced by the activity of multidrug efflux transporters, including *ABCC* and *ABCG2*, members of the *ABC* family [28,29]. Other general drug resistance mechanisms include alterations in cellular metabolism, signal transduction pathways, proliferative capacity or qualitative and/or quantitative alterations in the drug target [30,31,32]. 

In addition to these mechanistic studies focusing on a single target, a few studies describe genome wide gene expression profiles that were associated with drug resistance in pediatric AML patients. Lamba et al. [33], combined genome wide-gene expression data, ex vivo Ara C response data and clinical response parameters from 88 AML patients. Markers were identified that were predictive for beneficial (240 probe sets) or a detrimental (97 probe sets) Ara C related clinical response. McNeer et al. correlated clinical data with whole genome DNA and RNA sequencing in a set of 28 patients, and showed that certain recurrent molecular aberrations (e.g., *NUP98* rearrangements) are predictive for induction therapy failure, but did not observe distinct expression patterns in primary resistant pediatric AML patients [34].

The above data indicate that drug response of AML patients is (partly) due to intrinsic properties of leukemic cells and may be modulated via the expression of specific proteins. Detailed knowledge of the cellular mechanisms that determine overall drug response towards commonly used chemotherapeutics in AML is lacking, and remains crucial for the development of new treatment strategies. Here, we hypothesized that AML cells possess a priori usage of specific signaling pathways—which can be detected by gene expression profiles—that contribute to drug response. To explore this hypothesis, we studied the relationship between gene expression and ex vivo response to four chemo-therapeutic drugs in a discovery cohort of 73 pediatric AML patients. We observed genes related to previously reported pathways and novel genes to be related to ex vivo drug resistance. Five uncovered genes associated with response to multiple drugs were selected, and differential expression was validated in an independent cohort of 48 pediatric AML patients. Our gene-expression-based data show that the heterogeneous drug response genes relate to key pathways, which may underlie the molecular basis of cellular drug resistance in pediatric AML. With this data, we aim to contribute to the understanding of response mechanisms and the development of novel approaches to circumvent drug resistance for the individual patient.

## 2. Results

### 2.1. Patients and Ex Vivo Drug Response of Primary AML Blasts

A schematic overview of experiments and analyses is given in Figure A1. In the discovery cohort, 63.4% of patients were male, with overrepresentation of MLL-rearranged AML patients (27.5%, Table A1). We tested primary blast cells of de novo pediatric AML patients for ex vivo response towards the chemotherapeutic agents Ara C (*n* = 121), DNR (*n* = 119), cladribine (2-CdA) (*n* = 103) and VP16 (*n* = 70), which are used for the treatment of AML routinely, or in trial setting [35,36,37]. All drugs showed a dose-dependent cytotoxicity in the tested drug concentration ranges (representative dose-response curves are shown in Figure 1), with exception of one sample in the discovery cohort, and three samples in the validation cohort, which did not reach the LC_50_ with the used drug concentration ranges for one or more drugs (Appendix A). The LC_50_ values for all cytotoxic drugs from the discovery and validation cohorts are shown in Figure 2A,B and Table A2. Of note, a selection of patients in the validation cohort showed relatively high LC_50_ values for 2-CdA, which were not present in the discovery cohort. Ex vivo drug response varied among karyotype groups, with the most apparent differences observed towards 2-CdA (Kruskal-Wallis *p* = 0.063, Figure A2). Overall, no statistically significant differences were observed between karyotypes. We then investigated whether aberrations in recurrently mutated genes were associated with the observed drug response profiles. Although small subgroups hampered these analyses, ex vivo drug response was not significantly associated with recurrent gene mutations (unpaired *t*-test *p* ≥ 0.1, Appendix A and Table A2). Of note, samples with *CEPBA* mutations displayed a slightly enhanced drug response towards all drugs compared to wild type samples (*p* = 0.13, Appendix A). To test whether samples displayed ex vivo cross-resistance, we calculated the Spearman’s rank correlation using the LC_50_ values of the four drugs (Figure 2C). In the discovery cohort, a cross-resistance of patients’ blasts was observed between VP16 and the other drugs: Ara C (r = 0.38, *p* = 0.02) and in particular DNR and 2-CdA (r = 0.61 and r = 0.54, *p* < 0.001 and *p* = 0.001, respectively) (Figure 2C). Moreover, similar drug responses were observed within patients for the two nucleoside analogues, Ara C and 2-CdA (r = 0.44, *p* < 0.001). In contrast to others, who reported ex vivo cross-resistance towards Ara C and DNR using similar methods, we observed no correlations between LC_50_ of these drugs [18]. Although differences in the mechanism of action of the drugs may explain this, the possibility cannot be excluded that the lack of this cross-resistance is due to the relatively small sample size. Independent validation of these findings in 48 patients confirmed a cross-resistance between 2-CdA, VP16 and DNR, but not with Ara C (Figure 2D).

### 2.2. Correlation of Gene Expression with Ex Vivo Drug Response

To assess associations between gene expression levels and ex vivo drug response, we correlated microarray data with ex vivo LC_50_ values. For Ara C, DNR, 2-CdA, and VP16, we identified 128, 461, 279, and 731 probes, respectively, that were associated (*p* < 0.001) with ex vivo drug response, representing 101, 345, 206, and 599 unique coding genes. Figure 3 shows heatmaps of the intensity levels of all associated probes with a *p*-value < 0.001, with patients ordered according to ex vivo drug response (LC_50_) and annotated with gene mutations. Gene mutations did not demonstrate specific patterns associated with the observed gene expression profiles, although several samples with *CEBPA* and *WT1* clustered based on similar gene expression profiles were associated with DNR and VP16 response (Figure 3B,D). Distinct gene expression was especially apparent in patients with extremely low or high LC_50_ values. A complete overview is provided in Appendix A. Table A3 shows the top ten positively and negatively correlating genes for each drug. Although drug response data for VP16 were only available for a selection of samples, the strongest correlations between the ex vivo drug response and gene expression were observed for VP16, with the Spearman’s correlation coefficients ranging from r = −0.75 to 0.67 and FDRs between 0.004 and 0.073.

Considering the similar responses of primary AML blasts from individual patients towards the tested drugs, we hypothesized that certain gene expression profiles may correlate with an overall ex vivo drug response profile, including several drugs that have a distinct structure and mechanism of action. Therefore, we assessed the overlap between genes that correlated with ex vivo response data of the individual drugs (Appendix A and Figure 4). A total of 119 protein-encoding genes were shared by two or more of the drugs analyzed. Overlap of drug response-associated genes was most frequent for VP16 and DNR.

We then assessed whether the gene expression profiles and associated pathways could also represent any known mechanisms underlying response. Therefore, we performed Gene Ontology analysis (GO, to identify associated molecular and biological processes), Pathway analysis (to identify in which pathways these genes may be enriched), and upstream regulon analyses (iRegulon, to identify which transcription factors and/or motifs are predicted to be responsible for the observed gene expression patterns). For these analyses, we used the genes that correlated with ex vivo response to the four drugs (listed in Appendix A, summarized results in Table A3 and Table A4 and Appendix A, full results in Appendix A). 

The most comprehensive and strongest associations between gene expression and ex vivo response were observed in VP16 (731 probes). The gene expression was associated with GO categories and canonical pathways that are involved in metabolic processes, chromosome organization, cell cycle DNA replication, base excision repair, DNA repair, and mitotic cell cycle (Figure 5). The genes implicated in these cellular processes were strongly enriched, and are shown in the gene networks in Figure 5 and in Appendix A, and include, for example, *TOP2A*, *GTSE1*, *STAG1*, *BRCA1*, and *E2F8*. Most genes displayed lower expression in the patient samples that were relatively resistant to VP16. Upstream regulon analyses revealed *E2F7*, *TFDP3*, and *E2F8* as the most important transcription factors, with 229, 280, and 215 gene targets, respectively. Genes of the *E2F* family play a crucial role in cell cycle regulation and apoptosis upon treatment with chemotherapeutic drugs [38], with *E2F7* and *E2F8* being negative regulators [39]. *TFDP3* is a member of the DP transcription factor family which heterodimerizes with *E2F* to promote transcription of *E2F* target genes. *TFDP3* overexpression confers chemo resistance in minimal residual disease in childhood T-cell acute lymphoblastic leukemia [40]. 

Ex vivo Ara C response was associated with 128 probes, related to e.g., H3-K4 specific histone methyl transferase activity, according to the associations with elevated expression levels of *KMT2B*, *KMT2D*, and *RBBP5*. Gene ontology analysis demonstrated that these genes were mainly involved in epigenetic gene regulation. Gene regulatory analysis predicted that the observed gene expression profiles were associated with upstream regulation by *ZNF513* (20 target genes) and *ELF5* (16 target genes). *ELF5* is a transcription factor regulating key genes e.g., *FOXA1*, *EGFR*, and *MYC*, and has been described as potential regulator of anti-estrogen resistance in luminal breast cancer [41] and imatinib resistance in chronic myeloid leukemia [42].

GO and pathway analysis on genes that were correlated with ex vivo DNR response (*n* = 461) showed enrichment in canonical pathways that mediate RNA processing, DNA replication, and growth hormone signaling. *MEF2A* was predicted to regulate the observed gene expression patterns with 62 target genes. It has been reported that a member of the *MEF2* family, *MEF2C*, is associated with chemo resistance upon phosphorylation of the serine 222 residue in AML patients [43]. 

Among the GO categories and pathways that were associated with the gene expression patterns that correlated with ex vivo 2-CdA response (279 probes) were pathways and cellular functions involved in the modification, transcription, and binding of DNA and RNA. Other pathways were related to transcriptional deregulation in cancer and immune response. Upstream, *STAT3* was predicted to regulate expression levels of 56 of the associated genes. *STAT3* was also observed to be positively correlated with drug resistance towards DNR and 2-CdA in this data and has been described extensively as a regulator of pathways related to tumorigenesis, tumor growth, and drug resistance [44].

### 2.3. Technical and Independent Validation of the Association between Expression Levels of Selected Genes and Ex Vivo Drug Response

Next, we aimed to evaluate the validity of the microarray data analyses. Based on strong association with ex vivo towards multiple drugs resistance in the discovery cohort, we selected *CLEC7A* [45], *HIF1A* [46,47], and *RUNX1* [48] for technical validation in the discovery cohort. Moreover, considering their known role as predictor of survival in AML, and the observed association with ex vivo response in the discovery cohort, we additionally evaluated *STAT3* [44] and *BRE* [49] using qPCR for technical validation in 58 of the 73 patients from the discovery cohort. The correlation coefficients between microarray and qPCR data were statistically significant, with the best concordance observed for *CLEC7A*, *HIF1A*, and *BRE* (Figure 6A,B,E; Spearman’s r ranging from 0.65 to 0.88, *p* ranging from 9.5 × 10^−8^ to <2.2 × 10^−16^). In the discovery cohort, the expression levels of *HIF1A* and *CLEC7A*, based on qPCR data, correlated significantly with drug response to VP16 and DNR, respectively, with the strongest correlation between the DNR response and *CLEC7A* expression (r = 0.52, FDR < 0.001, Figure 6F). Expression levels of *BRE* were, as expected, inversely correlated to drug response to 2-CdA and VP16. It is important to emphasize that qPCR has a lower limit of detection, leading to an underestimation of the correlation for genes with many samples at these low ranges of expression. *RUNX1* and *STAT3* levels did not correlate with the LC_50_ values of any drug, which is possibly explained by the poorer concordance between qPCR and microarray data (r = 0.44–0.47, *p* = 6.4^−4^–1.8^−4^). 

To test validity of the data in an independent dataset, we performed qPCR for the abovementioned panel of five genes in the 48 pediatric AML samples of the validation cohort. The LC_50_ distribution in these samples towards Ara C, DNR, 2-CdA, and VP16 is shown in Figure 2B. As distinct gene expression profiles were especially notable in samples with very high or low LC_50_, we compared sensitive and resistant samples based on a cut-off at the 30th and 70th percentile. This cut-off was a trade-off between selecting truly different response subgroups (e.g., the 25th and 75th percentiles) and the appropriate number of patients per group for statistical analyses. Overall, expression differences between resistant and sensitive samples were subtle (Figure 7). *RUNX1* and *STAT3* were not differentially expressed among drug-response groups (Figure 7D and data not shown). Samples with high VP16 LC_50_ values had higher *CLEC7A* and *HIF1A* expression, compared to samples with LC_50_ values below the 30th percentile (Wilcoxon Rank Sum test, *p* = 0.03 and *p* = 0.02, respectively, Figure 7E,F). *HIF1A* expression was significantly higher, and *BRE* expression was significantly lower in DNR-resistant samples (Figure 7B,C, *p* = 0.004 and *p* = 0.01, respectively), in line with previous reports where high *BRE* expression was observed to predict favorable outcome in MLL-rearranged AML patients [49,50]. 

Although we primarily aimed to investigate ex vivo drug response, we addressed whether ex vivo response was associated with adverse clinical outcomes. In this relatively small pediatric AML-population, we did not observe such associations. Additionally, cross-resistance in the discovery cohort (defined by LC_50_ above the median for two or more drugs) was not predictive for adverse clinical outcomes. We investigated whether expression levels of genes associated with ex vivo drug response would have prognostic potential, using the AML dataset by Li et al. [51] and data from TCGA [52]. In the first dataset, *CLEC7A* expression was associated with a poorer overall survival compared to low *CLEC7A* expression (*p* = 0.0013, optimal cut-off at the 11th percentile). Additionally, low *BRE* expression (*p* = 0.021–<0.001, depending on selected probe) and high *HIF1A* expression predicted poor overall survival in this dataset (*p* = 0.0053, optimal cut-off at the 37th percentile). In the TCGA dataset, only high and intermediate *HIF1A* expression were predictive for shorter disease-free survival, compared to low expression, below the 30th percentile (*p* = 0.0071). Although this analysis lacks the combined effect of these gene expressions and multivariate validation, these data suggest the added value of the observed gene expression profiles associated with ex vivo drug response in the discovery set.

## 3. Discussion

In summary, we provide a transcriptomic dataset of primary pediatric AML samples in relation to ex vivo drug response towards Ara C, DNR, VP16 and 2-CdA. We show meaningfulness of the data by providing examples of known biology underlying ex vivo drug response towards chemotherapeutical drugs, as well as by providing internal and external validation data for selected genes (available through Appendix A). 

Among the relatively low number of genes associated with Ara C response, we observed the enrichment of genes related to epigenetic regulation (e.g., *KMT2B* and *KMT2D* (also known as *MLL2* and *MLL44*)). In pediatric AML, MLL genes are notorious because of the frequent rearrangements that comprise a cytogenetic subgroup of often poor prognosis patients [53]. Wild type MLL genes play a crucial role in eukaryotic transcription factor 2 (*E2F*)-mediated DNA damage response and apoptosis. *E2F1* associates with the MLL family of histone methyl transferases, and with host cell factor 1 (*HCF1*), to induce apoptosis upon DNA damage [54]. It has become increasingly clear that *E2F1*, through complex functions via the p73/DNp73-miR205 axis, is both involved in regulating drug resistance and cancer progression as a genotoxic-treatment induced apoptosis [41,55,56], with context-dependent effects [57,58]. We observed *E2F7* and *E2F8* as negatively correlated to drug resistance (Appendix A) and, as predicted upstream regulators of the observed gene expression profiles (Appendix A). This suggests that expression profiles may be the result of shared transcription factors and/or motifs related to *E2F*. Indeed, shared motifs regulating *E2F* family members were predicted to control the expression of genes correlated with both Ara C and DNR, as well as DNR and VP16 (Appendix A). The functional interpretation of these observations remains difficult, as conflicting results have been published on the role of over- and underexpression of these transcription factors in relation to cancer [59,60,61,62,63,64]. Although finding transcription factors related to cell cycle and apoptosis may seem logical, as most cytostatic agents target fast cycling cells, these observations both confirm the quality of the data and stress the function of this upstream regulator. 

The most frequent overlap of genes that correlated with drug resistance was observed for DNR and VP16. Interestingly, these genes are implicated in the mechanism of action of these drugs; for example, the gene expression levels of DNA polymerases *POLE3-4* were inversely correlated to DNR resistance. In concordance with previous reports [65], expression of the *TOP2A* gene—the direct target of VP16—was inversely correlated with ex vivo VP16 resistance (r = −0.57). Expression levels of members of the solute carrier family proteins (*SLC*) were associated with ex vivo resistance towards all antitumor drugs (Appendix A). *SLC* proteins comprise a large family of transporters, some of which are directly or indirectly associated with drug resistance. In this respect, *SLC2A3* (*GLUT3*) expression was positively correlated with drug resistance towards DNR, VP16, and 2-CdA (r = 0.44, r = 0.55 and r = 0.48, respectively, Appendix A). *SLC2A3* expression levels were found to be increased in virtually all types of cancer cell lines, including leukemia. In these cell lines, it was also shown that the use of glucose transport inhibitors that interact with *SLC2A3* and related molecules can sensitize cancer cells to drugs including DNR under hypoxic conditions [66].

In an independent validation cohort, we observed that *HIF1A*, *BRE*, and *CLEC7A* levels were related to drug response. The roles for *HIF1A* and *BRE* have been previously described in drug resistance and clinical outcomes in myeloid malignancies [49,51]. Although the *CLEC* genes, especially *CLEC12A* or *CLL-1*, have been previously linked to drug resistance, and proposed as treatment targets in AML [67], *CLEC7A* was not yet associated with a drug resistance phenotype. Specifically, *CLEC7A* plays a role in innate immune response, and is thus predominantly expressed in the bone marrow, lymphoid tissue, and blood. High *CLEC7A* expression has been identified as an adverse prognostic factor in renal cell carcinoma [45]. 

For the appropriate translation of our findings to the clinic, a major limitation of our study is the response measurements of single drugs, while in the clinic, combinations of chemotherapeutic drugs are common practice to overcome resistance in the heterogeneous AML cell populations. This may partly explain the lack of association with clinical outcomes. Moreover, differential expression in single genes may not be able to elicit a clear phenotype. In summary, we showed gene sets and signaling pathways relevant for ex vivo drug response. These results should be explored in more detail to allow further clinical translation. 

## 4. Materials and Methods

### 4.1. Patient Samples

The current study group of 121 pediatric AML patients was divided in a discovery cohort (*n* = 73), for which microarray-based gene expression data was available [68], and a validation cohort (*n* = 48). The study was approved by the Institutional Review Board according to national law and regulations and informed consent was obtained for all patients. Patients were diagnosed with AML between 1984 and 2017. Clinical patient characteristics are summarized in Table A1 and an overview of tests performed is given in Table A5. Viably frozen bone marrow or peripheral blood samples were provided by the Dutch Childhood Oncology Group (DCOG) and the ‘Berlin-Frankfurt-Münster’ AML Study Group (BFM-AML SG). RNA or cDNA samples for qPCR were provided by the DCOG or the Princess Maxima Center for Pediatric Oncology in Utrecht (PMC).

Leukemic cells were isolated by Ficoll gradient centrifugation (1.077 g/mL Amersham Biosciences, Freiburg, Germany) and non-leukemic cells were eliminated as previously described [20]. Detailed leukemic cell isolation procedures are available in the Appendix A. After processing, samples contained more than 80% leukemic cells, as determined by morphology, using cytospins stained with May-Grünwald-Giemsa (Merck, Darmstadt, Germany). A minimum of 5 × 10^6^ leukemic cells were lysed in Trizol reagent (Invitrogen, Life Technologies, Breda, The Netherlands). Genomic DNA and RNA were isolated as described [49]. Leukemic samples were routinely analyzed for cytogenetic aberrations by chromosome-banding analysis, and screened by participating childhood oncology groups for recurrent non-random genetic aberrations characteristic for AML, including MLL-rearrangements, inv(16), t(8;21), and t(15;17), using either RT-PCR and/or fluorescence in situ hybridization (FISH).

### 4.2. Ex Vivo Drug Response

Ex vivo cytotoxicity of the deoxynucleoside analogues 1-β-D-arabinfuranosylcytosine (Ara C, Cytosar; Pharmacia & Upjohn, Woerden, The Netherlands) and 2-chlorodeoxyadenosine (2-CdA, Leustatin, Ortho Biotech, Horsham, PA, USA), the anthracycline daunorubicin (DNR, Cerubidine, Rhône-Poulenc, Maisons-Alfort, France), and the topoisomerase II inhibitor etoposide phosphate (VP16, etoposide-TEVA; TEVA-Pharma, Haarlem, The Netherlands) was determined after 96 h drug exposure using MTT [69] on fresh, non-cryopreserved primary AML samples. Briefly, six concentrations of each drug were used in the following ranges: Ara C (0.04–41 μM); 2-CdA (0.001–140 μM); DNR (0.004–4 μM), and VP16 (0.09–3.4 μM). Cells without any drug added were included as controls and wells containing culture medium only were used as blanks. The cells were cultured for 96 h at 37 °C in a humidified atmosphere containing 5% CO2, after which 10 µl of 3-[4,5-dimethylthiazol-2-yl]-2,5 diphenyl tetrazoliumbromide (MTT; 5 mg ml-1, Sigma Aldrich, Zwijndrecht, The Netherlands) was added. Formazan crystals (indicating metabolically viable cells) were dissolved using acidified isopropanol (0.04 N HCl-isopropyl alcohol) and the optical density (OD) was measured spectrophotometrically at 562 nm and 720 nm. Importantly, only high quality data were used with stringent criteria, since 25% of ex vivo samples cannot be used in ex vivo assays, due to limited viability or low blast percentage, and 20% of MTTs are not evaluable, due to low number of remaining blasts at day 4. Evaluable results were obtained when a minimum of 70% leukemic blast cells was present at day 4 in control wells and when the control OD was >0.05. Dose response curves were obtained, and drug responses were summarized using the LC_50_ value, the drug concentration achieving 50% lethality of the leukemic cells. Details methods for the ex vivo drug response assays are available in the Appendix A.

### 4.3. qPCR

DNA and RNA samples were available from routine processing of diagnostic samples. cDNA was synthesized from RNA samples as previously described [70]. Probes for *RUNX1* (Hs00231079_m1), *HIF1A* (Hs00153153_m1), *BRE* (Hs01046283_m1), *STAT3* (Hs00374280_m1), *CLEC7A* (Hs01902549_s1), and *GAPDH* (Hs02786624_g1) (Thermo Scientific, Waltham, MA, US) were used to quantify gene expression, according to the manufacturers’ instructions and as previously described [70]. All gene expression data were calculated relative to *GAPDH* expression using the delta(CT) method.

### 4.4. Data Preprocessing and Statistical Analyses

Data were preprocessed as previously described [68]. The variance stabilization normalization procedure (VSN) [71] was applied to remove background signals and to normalize raw data across arrays. Log2 transformed expression values were calculated from perfect match (PM) probes only, and summarized using median polishing. The original and processed data have been deposited in the NCBI Gene Expression Omnibus (GEO; http://www.ncbi.nlm.nih.gov/geo) under GEO Series accession number GSE17855. Expression values were correlated with the LC_50_ values of the MTT assays using the Spearman’s rank correlation. A nominal *p*-value of 0.001 was used to identify correlated genes. *p*-value correction was performed using FDR approach. Group means were compared according to the appropriate statistical methods considering distribution of the data, and are indicated per situation in the results section.

### 4.5. Software

RStudio (version 3.6.1) was used to run the abovementioned analyses (RStudio Inc, Boston, MA, USA). Hierarchical clustering analysis with average linkage was performed and visualized using ComplexHeatmap [72]. Correlation analyses were performed using base R functions and visualized using corrplot [73]. Gene Ontology (GO) and Pathway analyses were performed using g:Profiler [74]. Upstream regulator analyses were performed using String (https://string-db.org/) and the Cytoscape (version 3.7.2) plugin iRegulon (version 1.3) [75,76,77].

## 5. Conclusions

Here, we provide novel gene-expression based data, supporting the notion that multiple deregulated pathways can lead to distinct drug response phenotypes in myeloid neoplasia [78]. Further studies are required to clarify whether or not modulation of the identified pathways may improve response to cornerstone chemotherapeutic drugs that continue to be our first line treatment of AML. Ultimately, we anticipate that this will support the development of novel clinical approaches, leading to deeper responses with elimination of the residual resistant cells, thereby preventing relapse and improving the prognosis of AML patients. 

## Figures and Tables

**Figure 1 cancers-12-01247-f001:**
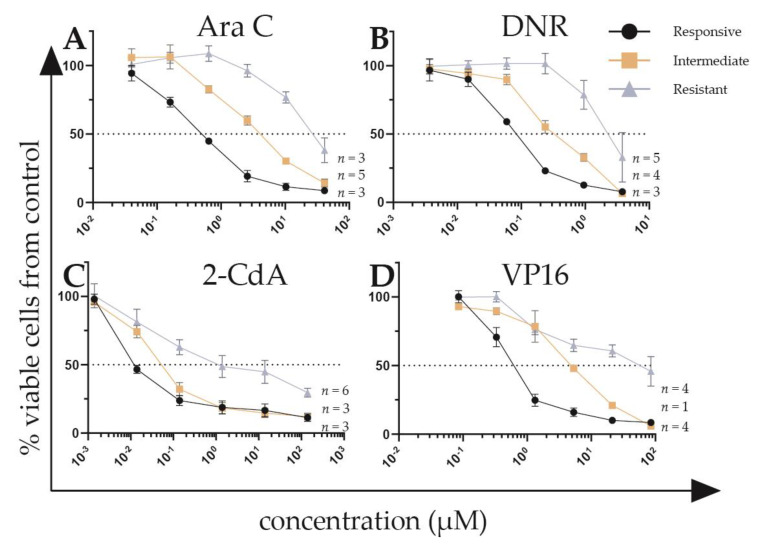
Representative dose-response curves for ex vivo drug response of primary acute myeloid leukemia (AML) samples towards (**A**) cytarabine (Ara C), (**B**) daunorubicin (DNR), (**C**) cladribine (2-CdA), and (**D**) etoposide (VP16). Responsive and resistant samples have LC_50_ values below the 30th and 70th percentile, respectively. Error bars represent standard error of the mean. The number of samples visualized is displayed per response-group. Data from each patient sample was assessed using technical duplicates.

**Figure 2 cancers-12-01247-f002:**
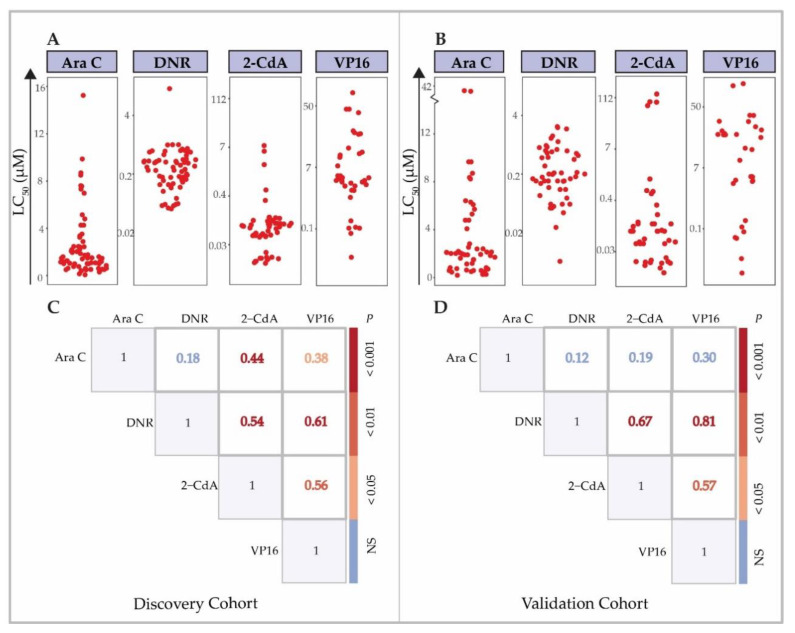
Distribution of ex vivo drug response data (LC_50_) for chemotherapeutic drugs and cross-resistance. ex vivo drug response of primary AML cells to: Ara C, DNR, 2-CdA and VP16. Data are depicted as LC_50_ values, the drug concentration (µM) at which 50% of the cells in the assay die. Each dot represents the LC_50_ value of an individual sample. (**A**) Ex vivo drug response in discovery and (**B**) independent validation cohort. Cross-resistance between the tested drugs in (**C**) the discovery cohort and (**D**) validation cohort. Numbers represent the Spearman’s correlation coefficient of LC_50_ values between shown drugs. *p*-value is calculated likewise and depicted according to the indicated color scheme.

**Figure 3 cancers-12-01247-f003:**
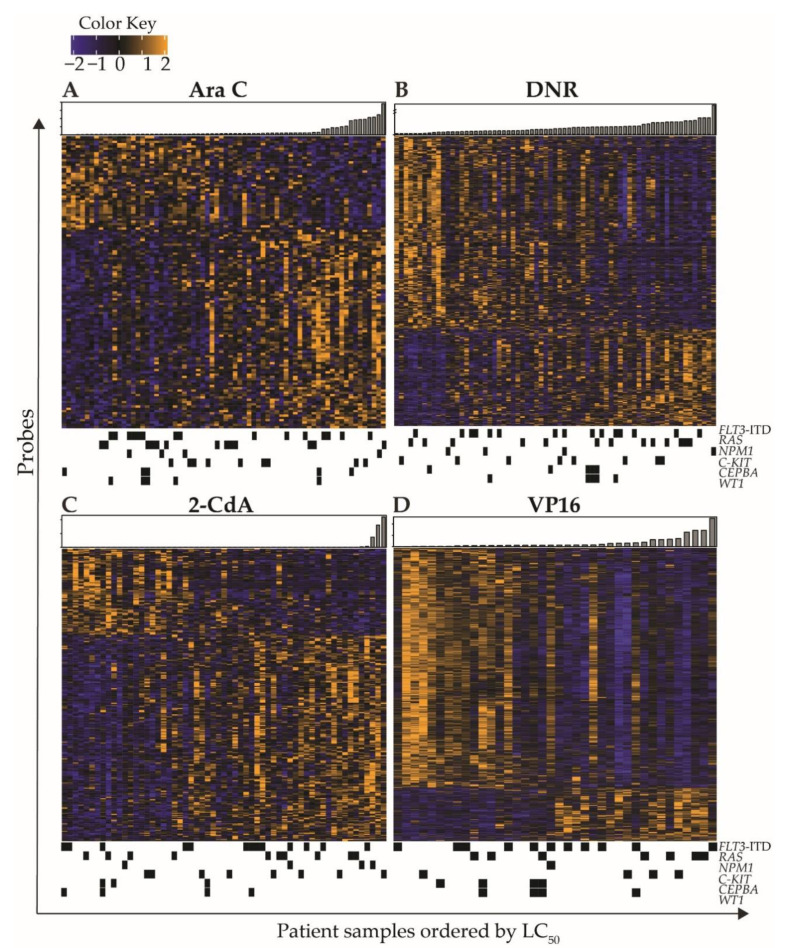
Heatmaps of ex vivo drug response associated gene expression patterns of pediatric AML blasts. Expression of probes was correlated to ex vivo drug response LC_50_ values using the Spearman’s Rank method. Probes were selected based on a *p*-value < 0.001. Probes (rows) were clustered using mean linkage, and samples were ordered according to ex vivo drug LC_50_ towards (**A**) Ara C, (**B**) DNR, (**C**) 2-CdA, and (**D**) VP16 LC_50_. Bar plots represent LC_50_ (μM) per patient. Black blocks underneath heatmaps indicate presence of recurrent mutations in samples.

**Figure 4 cancers-12-01247-f004:**
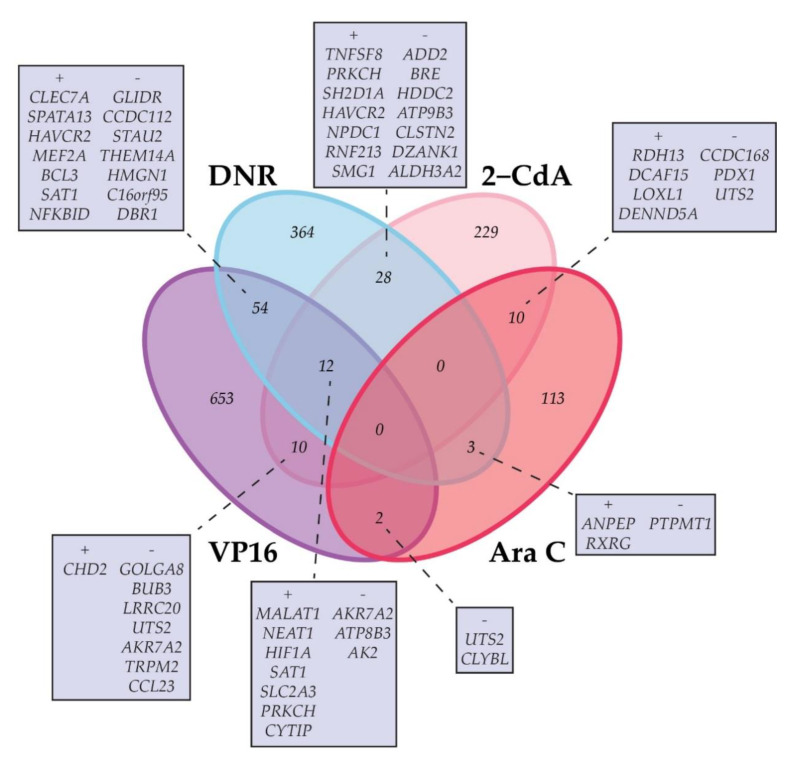
Schematic depiction of the number of genes that (inversely) correlate significantly (Spearman’s rank correlation *p*-value < 0.001) with ex vivo drug response in a single or two or more of the studied drugs (Ara C, VP16, DNR, 2-CdA). A + indicates a positive correlation with high LC_50_ values, indicating relatively resistant samples. A − indicates a negative correlation with resistance. Overlapping genes are shown, ranked according to the Spearman’s rank correlation coefficient. A maximum of 14 genes are shown. A complete overview of overlapping genes is given in Appendix A.

**Figure 5 cancers-12-01247-f005:**
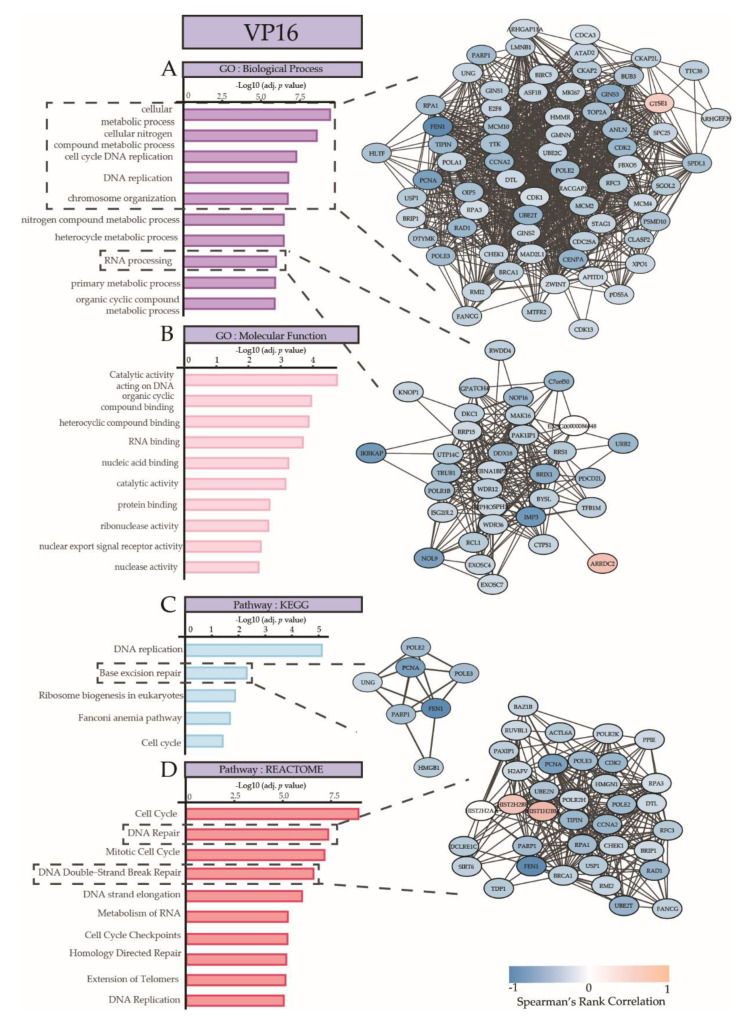
Summary of Gene Ontology Analysis and Pathway analysis on genes that are associated with ex vivo VP16 response, ranked according to adjusted *p*-values. (**A**) Biological processes and (**B**) molecular functions linked to genes associated with response towards VP16. Pathway terms from (**C**) KEGG and (**D**) REACTOME with significantly enriched genes that are associated with VP16 response. An interaction network visualization on the right depicts the gene–gene interactions for selected processes, pathways, or functions, indicated with dotted lines. Blue indicates a negative correlation and orange–red indicates a positive correlation with drug resistance.

**Figure 6 cancers-12-01247-f006:**
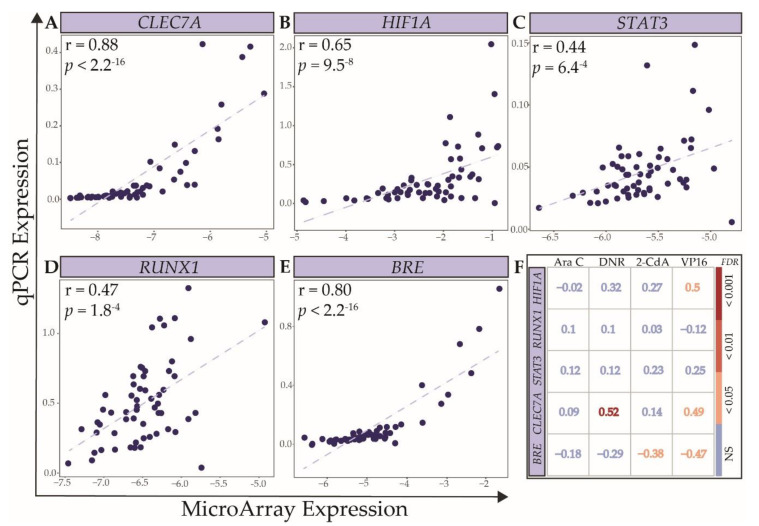
Technical validation of gene expression levels of selected genes. Scatter plots of gene expression levels measured by microarray and qPCR for: (**A**) *CLEC7A*, (**B**) *HIF1A*, (**C**) *STAT3*, (**D**) *RUNX1*, and (**E**) *BRE*. Correlation coefficients and *p*-values are calculated using the Spearman’s rank correlation. (**F**) The Spearman’s rank correlation coefficients and FDRs between LC_50_ of Ara C, DNR, 2-CdA, and VP16 and expression levels of the abovementioned genes, measured by qPCR in the discovery cohort.

**Figure 7 cancers-12-01247-f007:**
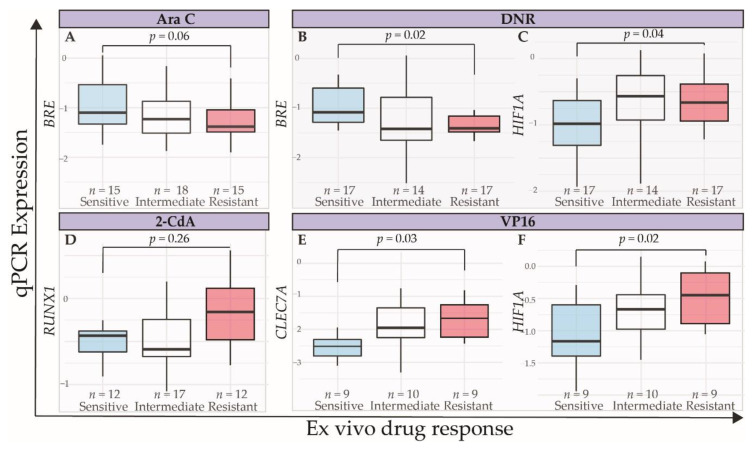
Independent validation of expression levels measured by qPCR of selected genes in samples of 48 pediatric AML patients. Expression levels of *BRE* in sensitive (light blue), intermediate (white), and resistant (red) patient samples, based on 30th and 70th percentile towards (**A**) Ara C and (**B**) DNR. (**C**) Expression levels of *HIF1A*, according to response to DNR. (**D**) Expression levels of *RUNX1*, according to response to 2-CdA. (**E**) Expression levels of *CLEC7A* and (**F**) *HIF1A*, according to the response to VP16. *p*-values were calculated using Wilcoxon Rank Sum test.

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
