# Peer review of "Harnessing Gene Expression Profiles for the Identification of Ex Vivo Drug Response Genes in Pediatric Acute Myeloid Leukemia"

_cancers, 2020, doi:10.3390/cancers12051247_

Round 1

Reviewer 1 Report

The authors have addressed my concerns.

Author Response

Reviewer #1: The authors have addressed my concerns.

Author reply: We are pleased to learn that we have addressed all concerns expressed by the reviewer. We thank the reviewer for the valuable comments concerning the previous version and believe that the quality of the manuscript has benefitted greatly from the revisions as proposed by the reviewer.

Reviewer 2 Report

This is already a revised manuscript. The original reviewers raised a number of concerns and questions, which were all adequately addressed and extensively answered. In addition, the authors also rewrote parts of the manuscript according to the reviewer's suggestions. 

Author Response

Reviewer #2: This is already a revised manuscript. The original reviewers raised a number of concerns and questions, which were all adequately addressed and extensively answered. In addition, the authors also rewrote parts of the manuscript according to the reviewer's suggestions. 

Author reply: We thank the reviewer for careful consideration of our revised manuscript and are pleased to learn that our responses to the concerns and questions regarding the previous version are deemed adequate.

This manuscript is a resubmission of an earlier submission. The following is a list of the peer review reports and author responses from that submission.

Round 1

Reviewer 1 Report

Harnessing gene expression profiles for the identification of ex vivo drug resistance genes in pediatric acute myeloid leukemia by Cucchi et al is an interesting study using microarray data from paediatric AML blasts that was published in 2011, coupled with ex vivo drug MTT assays to study the relationship between gene expression and response to four chemotherapeutic agents using a discovery (n=73) and validation cohort (n=48). Ex vivo, the authors showed a cross-resistance of patients' blasts between VP16 and the other drugs including Ara C and in particular DNR and 2-CdA. Independent validation of these findings in 48 patients confirmed cross-resistance between 2-CdA, VP16 and DNR. The study shows that VP16 resistance samples overexpress CLEC7A and HIF1A, compared to sensitive samples. HIF1A expression was shown to be expressed significantly higher, whereas BRE expression was shown to be significantly lower in DNR-resistant samples.

Major comments:

  • Analysis of the molecular function of gene expression data is interesting and well presented. I wonder why the authors have elected to include these data in the supplementary information rather than the main body of the manuscript. Further, clear information on whether activation or repression of these molecular pathways is associated with resistance to each of the chemotherapies would make the figure clearer. The expression profiles linked to VP16 resistance provides a plausible and understandable link to the resistance phenotype observed. As each of these molecular functions are critical for the survival the cells, I am wondering in each of these KEGG or REACTOME revealed p/w, were there any genes that may act in compensation (increased expression) to repair DNA damage, promote mitosis and the cell cycle that may form the targets of novel treatment strategies?
  • Caption or for Supplementary Table 1 needs more details.
  • Microarray data was generated in reference 51, in which 237 samples were available (GSE17855). In the current cohort, analysis of 73 of these samples was performed. More information of how ex vivo drug resistance studies were preformed using patients samples collected almost a decade ago would be appreciated.  AML samples are typically very difficult to grow ex vivo/in vitro (and in vivo for that matter), particularly following free-down? More clarity in the methods is required, or if not enough room in the main body of the manuscript more details should be included in a Supplementary Methods section.
  • It is not clear how data was combined for each resistance group, or the number of samples that fall into each group. Further, it is not clear how robust this analyses is? Was principle component analysis or similar performed to provide confidence in the grouping?
  • The authors talk about TFDP3 and E2F genes found to play a role in cell cycle and chemo resistance, however across the resistance profiles they both show decreased expression ?
  • Throughout the manuscript reference to correlated gene expression profiles fails to clearly indicate whether expression profiles in resistance subsets is increased or decrease. From what I can gather, the majority of candidates further investigated were for genes that negatively correlated with resistance?
  • Information on the statistical tests used and numbers of samples should be included in each of the figure captions. Figure captions are frequently above, rather than below figures.
  • The authors suggest that non-leukemic cells were eliminated as previously described in reference 16, prior to processing. Can the authors provide more detail on the elimination protocol performed as the original article referenced performed depletion to enrich for T ALL. Was this performed prior to free-down or subsequently on thaw for the drug screening and qPCR validation experimemts? It is of note that younger children were also investigated and included in the cohort Table A1. Patient characteristics, with acute monocytic leukemia (AML-M5) a common AML subtype in younger children and characterised by CD14 expression etc.
  • It is inappropriate to introduce new data in the discussion.

Minor grammar issues:

  • Some issues with line number formatting: 37-39, 95
  • Information of the specific number of samples in each drug-response group should be included in Figure 1.
  • Line 59: Intrinsic properties of leukemic cells that are thought to play an important role in resistance to Ara C concern, for example, the ability to transport the drug across the plasma membrane in both directions[17]. The sentence might better be structured: An intrinsic property of leukemic cells that is thought to play an important role in resistance to Ara C, is the ability to transport the drug across the plasma membrane in both directions[17].
  • Line 80: Detailed knowledge of the cellular mechanisms that are involved in overall drug resistance towards commonly used chemotherapeutics in AML is lacking and remains crucial for the development of new treatment strategies that overcome drug resistance, eradicate all leukemic cells with initial chemotherapy and ultimately reduce side effects and improve outcome in pediatric AML patients. The sentence might better be structured: Detailed knowledge of the cellular mechanisms of overall drug resistance towards commonly used chemotherapeutics in AML is lacking and remains crucial for the development of new treatment strategies. These data may help to develop strategies that overcome drug resistance, and eradicate all leukemic cells with upfront chemotherapy, which may ultimately reduce side effects and improve outcome in pediatric AML patients.
  • Line 97: Figure 1A should read (Supplementary Figure S1). I am not sure why the manuscript includes appendixed data rather than clear supplementary information?
  • The resolution of most of the Figures is very poor.
  • Tables A1, - A5 captions are located below the tables?

Reviewer 2 Report

In their manuscript ” Harnessing Gene Expression Profiles for the Identification of Ex Vivo Drug Resistance Genes in Pediatric Acute Myeloid Leukemia”, Cucchi et al. use MTT-based cell proliferation inhibition assays in primary diagnostic paediatric AML samples to correlate micro-array-based gene expression with experimental drug response datas. The authors on the one hand present some evidence of cross-resistance for 4 different AML-directed drugs, on the other hand, show surprisingly little overlap in the actual gene correlation data. While the hypothetical reasoning to suspect differential gene expression as a culprit for intrinsic chemotherapy resistance is sound, methodological shortcomings make the presented data difficult to interpret.

Minor comments:

In the introduction, the authors suggest “complete eradication of all leukemic cells” with initial therapy. They should mention the possibility of maintenance therapy as an alternative approachs that gains more interest with the advent of targeted therapies (e.g. FLT3 inhibitors).

It is rather surprising that the authors mention ara-C resistance as a major cause for relapse – without referring to a single recent publication from the last four years that focused on ara-C resistance in AML. Resistance mechanisms for ara-C are furthermore mentioned with a strong focus on transporters whereas well-characterised resistance mechanisms (like NT5C2 mutations, DCK downregulation and SAMHD1 overexpression) have not explicitly been mentioned at all.

Fig. 1: Concentrations should (also) be provided in a molar metric.

Resistance mechanisms in a dominant clone at diagnosis – relapse might be caused by resistant subclones – discuss!

Major comments:

LC50 might be misleading as plateaus in toxicity might be reached – in other words, LC50 values (as seen in Fig. 1) are not an appropriate measure. For VP-16, it seems that the half-maximal inhibitory effect is almost identical for “intermediate” and “resistant”. The authors should therefore provide a measure that reflects the cytotoxic kinetics. This is complicated by the fact that a single read-out for cell-viability (MTT assay) was used. It would have been informative to see whether other read-outs (e.g. ATP-relase assays) would have yielded similar results. The referee hypothesizes that those outliers (e.g. LC50 >> half-maximal inhibition) account for the “cross-resistance” – it is therefore of utmost importance to exclude artifacts stemming from the readout MTT.

It is striking that least “cross-resistance” is seen between ara-C and daunorubicin. This should be discussed in more detail.

It is unclear why the authors chose to validate their findings in the adult TCGA cohort – when there is a paediatric AML cohort (TARGET) readily available. One might suspect that those findings could not be replicated in the paediatric cohort.